# Global Warming, Advancing Bloom and Evidence for Pollinator Plasticity from Long-Term Bee Emergence Monitoring

**DOI:** 10.3390/insects12050457

**Published:** 2021-05-16

**Authors:** James Cane

**Affiliations:** 1Emeritus, USDA-ARS Pollinating Insect Research Unit, Utah State University, Logan, UT 8432, USA; jim.cane2@gmail.com; 2WildBeecology Consulting, Logan, UT 84341, USA

**Keywords:** phenology, seasonality, Anthophila, climate change, global warming, weather, ground-nesting, pollinator

## Abstract

**Simple Summary:**

Common experience has it that natural seasonal events are earlier in some years and later in others. In temperate zones, these events often seem to be timed by some combination of winter chill hours followed by cumulative warmth, variables which vary with a year’s weather. Over the course of decades, monitored annual bloom dates of a variety of wildflower species varies over 4–8-week ranges. This study reports annual emergence dates of four wild species of ground-nesting bees recorded over 12–24 years. Their first emergence ranged over 4–6 weeks, comparable to wildflowers and in part relatable to the same thermal cues used by plants. Global warming is advancing seasonal events, notably flowering, but it appears that native bees have the phenological flexibility to maintain their floral associations and critical pollination services both in wild and agricultural settings.

**Abstract:**

Global warming is extending growing seasons in temperate zones, yielding earlier wildflower blooms. Short-term field experiments with non-social bees showed that adult emergence is responsive to nest substrate temperatures. Nonetheless, some posit that global warming will decouple bee flight and host bloom periods, leading to pollination shortfalls and bee declines. Resolving these competing scenarios requires evidence for bees’ natural plasticity in their annual emergence schedules. This study reports direct observations spanning 12–24 years for annual variation in the earliest nesting or foraging activities by 1–4 populations of four native ground-nesting bees: *Andrena fulva* (Andrenidae)*, Halictus rubicundus* (Halictidae)*, Habropoda laboriosa* and *Eucera* (*Peponapis*) *pruinosa* (Apidae). Calendar dates of earliest annual bee activity ranged across 25 to 45 days, approximating reported multi-decadal ranges for published wildflower bloom dates. Within a given year, the bee *H. rubicundus* emerged in close synchrony at multiple local aggregations, explicable if meteorological factors cue emergence. Emergence dates were relatable to thermal cues, such as degree day accumulation, soil temperature at nesting depth, and the first pulse of warm spring air temperatures. Similar seasonal flexibilities in bee emergence and wildflower bloom schedules bodes well for bees and bloom to generally retain synchrony despite a warming climate. Future monitoring studies can benefit from several simple methodological improvements.

## 1. Introduction

Native bee faunas and wildflower communities continue to suffer from diverse human impacts, particularly in temperate zones of the Northern Hemisphere. Some trends are long-running, such as the conversion of flower-rich prairies and other arable wildlands to agricultural monocultures. Other disruptive factors are intensifying, including urban sprawl and global warming. A warming climate is leading to hotter, longer summers, and milder, shorter winters, thereby advancing the regional onsets of spring and summer. As a consequence, some plant species are blooming earlier in recent decades [1]. The most detailed records come from rare, long-term annual surveys of wildflower blooming schedules at a fixed location [2,3]. Several short-term manipulative field studies with several solitary bees show that warmer nesting substrates also elicit earlier adult emergence [4,5,6].

The season(s) of each bee species’ period of regional adult activity can be compiled from collection dates on pinned museum specimens. Records for abundantly collected species can also bracket the general calendar dates of seasonal activity, revealing latitudinal trends [7]. This has been a popular means to assess responses of bee communities to global warming (refs in [7]). These are averages, however, the bulk of the specimens coming from the height of the nesting season, not its onset. For this reason, they fail to capture local year-to-year variability in bee emergence, lending a false sense of uniformity to the start of the annual activity schedule for any given bee species, as well as exaggerating the duration of a species’ annual flight season. Bee populations are mostly challenged by years of abnormal weather events at their locale, a response that cannot be captured by retrospective analyses of specimens in collections.

If a warming climate is to dissociate bee emergence from host flowering, then as a corollary, bee emergence schedules should be less responsive/flexible with regard to the same seasonal cues that trigger flowering or they should be responding to different cues. For many bee and flowering species, the likely cues are either rainfall (deserts) or thermal, but experimental evidence is in fact uncommon not only for bees [4,5,6] but also for their perennial floral hosts. The blueberry *V. ashei* Reade is an instructive exception. Manipulative experiments and modeling revealed that its leaf flush and flowering required plants to first accumulate sufficient winter chill hours (centered around 0 °C). Only after plants had satisfied chill hour requirements did they then respond to accumulating growing degree days (GDD) warmer than a 7 °C baseline [8]. Studies of several *Osmia* bee species report a broadly similar thermal strategy driving wintering and subsequent spring emergence [9,10,11]. What has been lacking is documentation of natural variability in bees’ annual emergence schedules, which requires multidecadal field data at fixed locations.

The purpose of this long-term study was to monitor annual emergence schedules of four species of resident wild ground-nesting bees (Appendix A). The four bee species occur in three different biomes (Appendix A). They represent three bee families, two seasons of adult activity, and a range in floral specialization from oligolectic (taxonomic pollen specialist) to polylectic (floral generalist). The bees’ fluctuating emergence dates are compared with available air and soil temperatures drawn from compiled data recorded at nearby weather stations.

## 2. Materials and Methods

The onsets of adult activity were annually monitored (with a few gaps) for four ground-nesting bee species. (Table 1). The three solitary bee species are univoltine (one annual generation). The four species and locales are listed in Table 1 and as follows. ***Halictus rubicundus*** Christ. This primitively eusocial bee was tracked for annual nest initiation by wintering gynes occupying several nesting aggregations on the campus of Utah State University or nearby (41.7° N × 111.8° W, 1460 m). ***Eucera (Peponapis) pruinosa*** (Say). First summer activity of this *Cucurbita* oligolege was annually tracked at flowering squashes at a market garden in adjoining North Logan, Utah USA (41.8° N × 111.8° W, 1397 m). ***Habropoda laboriosa*** (Fabr.). First sightings of males and females were recorded as they visited this oligolege’s blueberry pollen hosts (cultivated *Vaccinium ashei* Reade, wild *V. elliottii* (Chapman) Small) growing southeast of Auburn, Alabama (32.5° N × 85.4° W, 187 m) [12]. After several years, a nesting aggregation of *H. laboriosa* was discovered, the nests hidden under leaf litter in the neighboring hardwood forest. That activity is not reported here. ***Andrena fulva*** Müller. This bee’s annual emergence dates were reported by Dr. Jozef Banaszak [13] at their nesting aggregation in a small wooded park in the city of Poznan, Poland (52.4° N × 16.9° E, elev. 69 m). Nesting sites of all four species were flat and fully sunlit during the nesting periods. Both the Polish and Utah sites experience subfreezing winter temperatures, with Logan Utah having longer, more severe winters with often substantial snowpacks.

To detect first bee activity, sites were inspected daily or every second day during favorable weather beginning before each bee species’ onset of activity. In Alabama, oligolectic *H. laboriosa* were monitored at ½ ha of cultivated *V. ashei* grown near their nesting site, as well as at nearby wild *V. elliottii,* whose bloom commenced weeks earlier (often in January). Both blueberries are primary pollen hosts of *H. laboriosa* [12]. Males and females were readily discerned on the wing. Oligolectic *E. pruinosa* were surveyed at flowers of 400–500 *C. pepo* summer squash plants grown annually in the same market garden in North Logan Utah. For both settings, local host bloom always preceded bee activity in all years. Like many oligoleges, the species are protandrous, with males first searching pollen hosts for receptive females [12]. For primitively eusocial *H. rubicundus*, gynes (potential reproductives) and males emerge and mate in the autumn. Only the gynes then overwinter somewhere away from their natal aggregation [14]. They return to their aggregation to nest the following spring. In a given year, as many as five of the known *H. rubicundus* nesting aggregations were active, depending on the discovery of new sites and loss of existing sites to landscaping practices or construction. All tracked *H. rubicundus* aggregations were at the same elevation and within 1.3 km of both each other and the recording weather station. Four nest sites were found on the campus of Utah State University, but the longest record was for an aggregation amid rock cobble at a neighborhood site (1998–2021).

Accessory instrumental and bloom data were collected for the two bee species studied in Logan Utah. First spring bloom of *Iris versicolor* was recorded at a garden 200 m from the neighborhood nest site of *H. rubicundus*. It is visited by the bee, and proved to be reliable, blooming early enough to roughly coincide with *Halictus* emergence (which often preceded bloom of even the earliest flowering ornamental shrub, *Forsythia*). To better evaluate temperatures experienced by pre-nesting bees, air and sunlit soil temperatures (5 cm depth) were manually taken at this bee’s campus cemetery aggregation on 5 and 17 March 2013; in that year, the first *Halictus* arrived on-site March 19. This same aggregation was earlier studied for a surface trait preferred by nesting females [15]. Local air temperatures and cumulative growing degree days (GDD) were taken from the USU campus weather station as compiled by the National Weather Service (https://w2.weather.gov/climate/xmacis.php?wfo=slc, accessed on 2 April 2021). The two available base temperatures of 10 and 18 °C were used for GDD summations for spring *Halictus* and summer *Peponapis*, respectively. Two-week blocks of hourly air temperatures were compiled from weather station data available at https://mesowest.utah.edu (accessed on 2 April 2021). Daily summer soil temperatures at 20 cm depth (approximate cell depth for *E. pruinosa*) were available and compiled for the years 2005–2020 from the NRCS Soil Climate Analysis Network (SCAN) records for station 2136 at Cache Junction (1349 m) 18 km across the valley from Logan. Linear regressions and other statistics using emergence and weather variables were calculated, and their assumptions checked (e.g., normality and constant variance) (SigmaPlot 12.5, Systat Software). Scatter plots were generated to visually evaluate prospects for meaningful correlations.

Published long-term phenological records of observed first bloom in plant communities are exceedingly rare, especially those with accessible raw data needed for calculating ranges in flowering dates. Two remarkable field studies were chosen. One comes from the Mohonk Reserve along the Hudson River valley of New York, USA [3]. From this study, first bloom dates of eight perennial spring wildflowers were chosen (Appendix A). They represent eight different families and had the most complete annual records for 1970–2006, roughly the time period and span for spring bees reported here. They are mostly herbaceous perennial wildflowers growing beneath a hardwood overstory. The other data set was provided by Dr. David Inouye for 1975–2000 from his high montane meadow site at RMBL in the Rocky Mountains near Gothic Colorado USA (archived at www.rmbl.org, accessed on 2 April 2021). He suggested annual dates of first bloom for three species attractive to native bees that together span the flowering season there: *Mertensia fusiformis*, *Delphinium barbeyi* and *Heterotheca villosa* (Appendix A). First spring bloom of some other species was missed when snowpack blocked access to RMBL. At that site, final snowmelt strongly influences blooming schedules [16]. The growing seasons at Mohonk, Poznan and Logan are comparable in length, and much longer than at Gothic.

## 3. Results

Annual bee emergence varied widely by calendar date (or day length). Among the four bee species, the annual date of first activity (Table 1) ranged over a period of 24–44 days. In sharp contrast, the local aggregations of *H. rubicundus* became active within the same 1–2-day period for any given year.

### 3.1. Halictus rubicundus

No single available measure accurately predicted the annual onset of this bee’s adult activity. For more than half the years, females were first seen at the neighborhood aggregation within six days of the year’s first air temperatures >15 °C. In 2013, 14 and two days prior to emergence at the cemetery site, the aggregation’s sunlit soil temperatures (5 cm depth) far exceeded ambient air temperatures (10.3 vs. 0.4 °C and later, 24.4 vs. 13.3 °C), illustrating the dramatic contribution of solar soil warming to the bee’s likely perception of seasonal warmth. This measure is not available from most weather stations, including the campus one.

For the neighborhood aggregation, the bees’ first arrival was poorly predicted by several seemingly relevant seasonal measures recorded at the nearby campus weather station. These include the annual date of final snowmelt (*r*^2^ = 0.16) and the date of the year’s last spring frost (*r*^2^ = 0.09). The annual date for first bloom of nearby *I. versicolor* also failed to predict *H. rubicundus* emergence (*r*^2^ = 0.19) although it corresponded closely in about half of the years (Figure 1). However, at one extreme, 2011 was the year of the latest spring killing frost (May 30) and was also the year of the latest emergence dates for both *H. rubicundus* (Figure 1) and, three months later, *E. pruinosa* (Figure 2).

Earliest nest initiation by *H. rubicundus* was partly predicted by a given year’s cumulative GDD up to the median date of first emergence (March 29) (*r*^2^ = 0.32, *p* ≤ 0.006). Degree days accumulated by that date can indicate the comparative warmth of spring in a given year (Figure 3). Thus, bees emerged early–all in March–for the five years with the warmest springs (cumulative GDD > 33 by March 29). For the five coldest springs, they emerged later in April (cumulative GDD < 0 by March 29). Cumulative GDD up to each year’s actual emergence date did not predict that year’s emergence date (*r*^2^ = 0.02). However, the range in annual cumulative GDD for actual emergence dates was nearly 4-fold less variable than if bees had emerged each year on the median emergence date of March 29 (variance = 91 vs. 324, *n* = 22 years) (Figure 3).

### 3.2. E. (Peponapis) pruinosa

The first sighting of this summer bee at squash flowers (Figure 2) corresponded to both air temperature GDD and summer soil temperatures. On average over the 20 year period, 396 GDD had accumulated by August 1st, a date when these bees were always nesting. In this period’s coldest summer, 2011, only 335 GDD had accumulated by August 1st. That was the latest recorded emergence year for both *E. pruinosa* (Figure 2) as well as for *H. rubicundus* months earlier (Figure 1). The soil temperatures that nesting squash bees would experience (recorded from across the valley) were likewise coolest in 2011 (19 °C) (average 22 °C) (Figure 4). In contrast, the years of earliest emergence (2016–2018 in Figure 2) also had the warmest valley soils by August 1st (24 °C each year).

### 3.3. H. laboriosa

Males emerged before females for all 12 years (protandry) (Figure 5). Emergence dates of males and females annually coincided with each other (*t* = −4.3, *p* ≤ 0.001), such that both sexes together emerged earlier or later in a given year. Although emergence of the first male *H. laboriosa* often preceded bloom of cultivated *V. ashei,* bloom at wild *V. elliottii* always preceded visits by the first males. Accumulated GDD from January 1st to emergence was not predictive of emergence for either sex (*r*^2^ < 0.01) unless one anomalous year (1992) was dropped from male emergence data (*r*^2^ = 0.32, *p* ≤ 0.07). The nearest weather stations lacked soil temperature sensors.

### 3.4. A. fulva

This European bee also flies in early spring, first emerging across a range of calendar dates over the recorded years (Figure 6) [13]. It is the only example here for which true emergence dates could be observed. No corresponding weather data are available to permit temperature interpretations.

## 4. Discussion

Schedules for the annual onset of adult activity for these four ground-nesting bees were only coarsely related to calendar date. The range of 24–44 days in these bees’ annual emergence dates (Table 1) is comparable to the overall life spans of active adult solitary bees. This span of dates is similar to those recorded for annual first bloom by perennial forbs during long-term monitoring at two U.S. locations (Appendix A), being the eight perennial spring forbs from the Mohonk Reserve (25–38 d) [3] and the 50–56-d range of three montane wildflowers growing near Gothic, Colorado. These four wild bees showed flexibility equal to that of perennial wildflowers in their responses to the variable weather of the north temperate zone, supporting a tentative conclusion that associations of plants and pollinators so far seem resilient to seasonal advances driven by global warming [17].

Shorter monitoring periods could give a skewed impression of trends and averages in annual bee emergence. Thus, emergence schedules of both *H. rubicundus* and later-flying *E. pruinosa* in Utah were earlier than average after 2012, but this period was immediately preceded by 4–5 years of tardy emergences (Figure 1 and Figure 2). A comparable shift to early emergence was apparent with *A. fulva* in Poland (Figure 6), but that was for >2 decades earlier. Depending on the time period, a 3–5 year monitoring study of these bees could have concluded that annual emergence varied little, is rapidly advancing, or is even lagging (as exemplified with *H. laboriosa*) (Figure 5).

Among the four ground-nesting bee species studied here, only the Logan population of *E. pruinosa* exists at the climatic extreme of its geographic range. This squash bee’s native floral hosts are perennial *Cucurbita* species found growing 1000 km further south in the far warmer U.S. desert Southwest. This bee only entered Utah about 170 years ago when European settlers began growing annual squashes [18]. Within the western range of *E. pruinosa*, the Logan population endures particularly long, cold winters and brief, cool summers, although these have eased in recent decades (Figure 4 and cited climate data sources) [18]. Nonetheless, its annual emergence in Logan was as flexible and versatile as those of the other three bee species (Table 1, Figure 2). Even the year of their latest emergence (2011) did not incur a reproductive cost, judging from their abundance at flowers the following year (433 bees) which exceeded their 12 year median abundance at the market garden (Cane, unpublished data).

The four monitored bee species not only tolerated annual variations in seasonal temperatures, but broadly responded to them. Annual emergence of *H. rubicundus* corresponded with cumulative GDD, the gynes emerging earlier in the warmest springs and later during chilly springs. Their responsiveness to temperature is indicated by the 4-fold lesser variance in GDD at actual emergence as compared with that of the median date of emergence over 22 years. Calendar date alone was a weaker predictor of their annual emergence than was GDD. These GDD values were generated from air temperatures recorded at a nearby weather station, which served as an imperfect surrogate for the more relevant soil temperatures experienced by these wintering bees near their nesting sites.

Temperature-driven phenological adaptability of non-social bees is consistent with the few field studies that shifted adult emergence schedules and nesting by manipulating thermal inputs to bees’ nesting substrates. Emergence of *Dasypoda hirtipes* (Fabr.) (*=plumipes*) was advanced by heating its nesting soils in late winter [5]. Likewise, the summer bee *Nomia melanderi* Ckll. emerged days earlier from plots in their nesting aggregation that were covered by squares of black plastic [4]. Conversely, that bee’s emergence was retarded by dusting soil surfaces with reflective white chalk dust [4]. Emergence of the cavity-nesting bee *Osmia ribifloris* Ckll. was briefly delayed by applying reflective white paint to their drilled nest blocks [19]. That study was at the hottest margin of that species’ range. Consequently, black-painted nest blocks disrupted post-winter emergence and enhanced mortality, probably because wintering bees were inadequately chilled by a winter that was too mild and brief. This outcome illustrates the limits of a bee species’ thermal tolerances during winter dormancy independent of lethal summer heat. The combination will probably cause many bees’ geographic ranges to withdraw from the hottest margins of their ranges as global warming intensifies. This retreat will accelerate a slower ongoing Holocene trend of poleward range shifts evidenced by other plant and animal taxa that have left a fossil record (e.g., [20]). Deleterious thermal effects of global climate change will likely manifest for bee populations living at the lowest elevations and latitudes of their species’ range. 

Some bee species respond to geographic variation in climate through additional adaptive life history traits. Some but not all bee taxa have versatile voltinism, adding more generations in hotter climes (e.g., [21,22]). As is evident from museum collection records, many others lack such versatility, including the four species monitored in the current study. Where populations of a bee species with a single fixed annual generation encounter prolonged hot growing seasons, they may instead employ an energy-saving prepupal diapause during the summer (e.g., [23]), a trait that is surprisingly difficult to detect. At the other climatic extreme, some alpine bee populations can even manage a two-year life cycle to accommodate their brief chilly growing season [24], but so far, few temperate-zone bee species show this capability. Lastly, alpine populations of some eusocial species, particularly *H. rubicundus,* can forego sociality where the brief growing season requires it [25]. At more local scales, flowering schedules can differ predictably with local microclimate even in relatively flat terrains [26]. In contrast, annual emergence at *H. rubicundus* local nesting aggregations in Logan was closely synchronous, no doubt reflecting the bee’s choice of similarly flat sunlit nesting sites surfaced with pebbles or cobble [15]. One exceptional study [27] found that emergence of a coastal dune-nesting bee and flowering by its primary floral host co-varied and were synchronized across five local nesting aggregations and adjoining plant populations living in a cool maritime climate. In general, comparable insights are rare for any given bee species simply because multiple natural nesting sites are rarely found near each other for any one species. 

Several practical features proved essential for this bee monitoring protocol. Because sites must be checked almost daily for several weeks during good weather, sites must be convenient to visit. For instance, aggregations of *H. rubicundus* were along my daily bicycle route to campus. Where possible, several sites should be found and monitored, as few places around a campus or town remain undisturbed over the course of several decades. Oligolectic bees are generally more suitable that polyleges for detecting emergence by monitoring host flowers. The broad host range of a polylectic bee necessitates inspecting too many potential floral hosts, excepting those seasons when few plant species are blooming, such as earliest spring. Lastly, choosing a bee species that one can learn to recognize on the wing at the nest site or floral host will simplify monitoring.

As surveys progressed, it became evident to me that my phenological criteria could be improved upon. In some years, emergence of the very earliest spring bees was immediately followed by a week or more of cold inclement weather. Did these early bees survive, or were they unfortunate outliers? A more representative measure of successful population emergence is desirable, such as the first date on which several individuals are seen on several successive days. Tracking peak emergence would be ideal, but it is unknowable until the season is past, plus its monitoring would entail hours of laboriously distinguishing and counting inconspicuous emergence holes or nest tumuli (soil heaps). Tumuli themselves are often indistinct and readily lost to rain and wind. Secondly, a measure of nearby bloom that roughly times with the onset of bee activity would be helpful, as bloom can serve as an integrator of various weather parameters (degree day accumulation, air and soil temperature, snow melt, etc.). As with emergence, the very first blossoms of a species may be spurious [2]. Instead, one should adopt an unambiguous rule for recording early bloom, such as the date when the first ten flowers open, or for shrubs, when half of the plants have a first flower.

## Figures and Tables

**Figure 1 insects-12-00457-f001:**
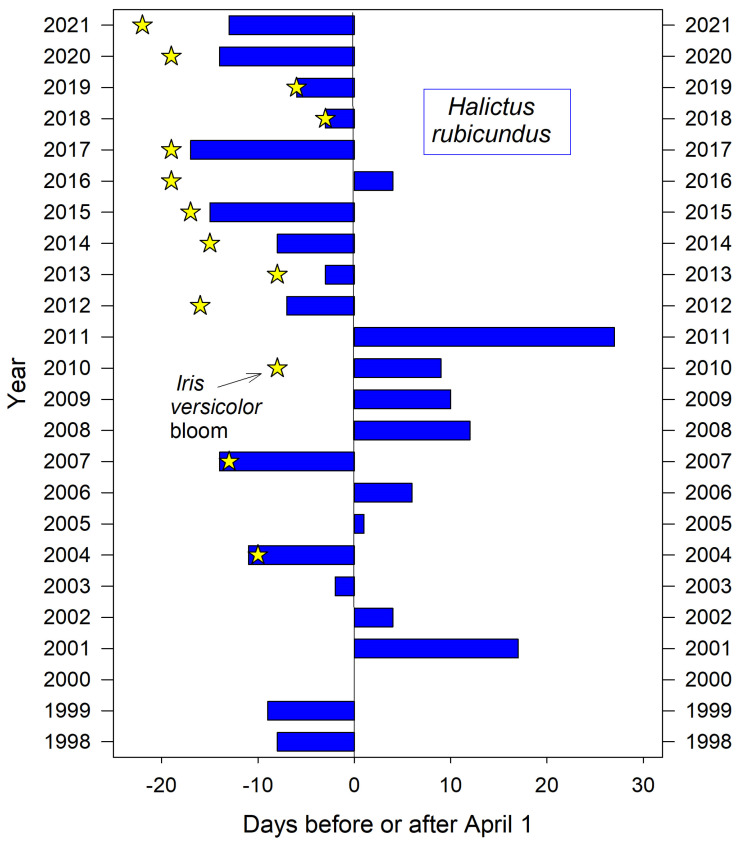
Variation in the annual date of first activity by *H. rubicundus* at their natal nesting aggregation in Logan Utah USA.

**Figure 2 insects-12-00457-f002:**
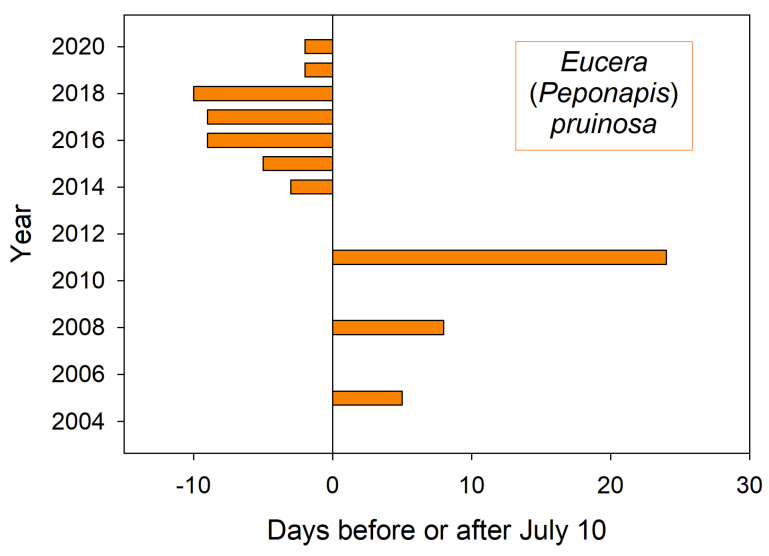
Annual date for first sighting of *E. pruinosa* at squash flowers (*Cucurbita*) in a market garden near Logan Utah USA. Missing years have no bar.

**Figure 3 insects-12-00457-f003:**
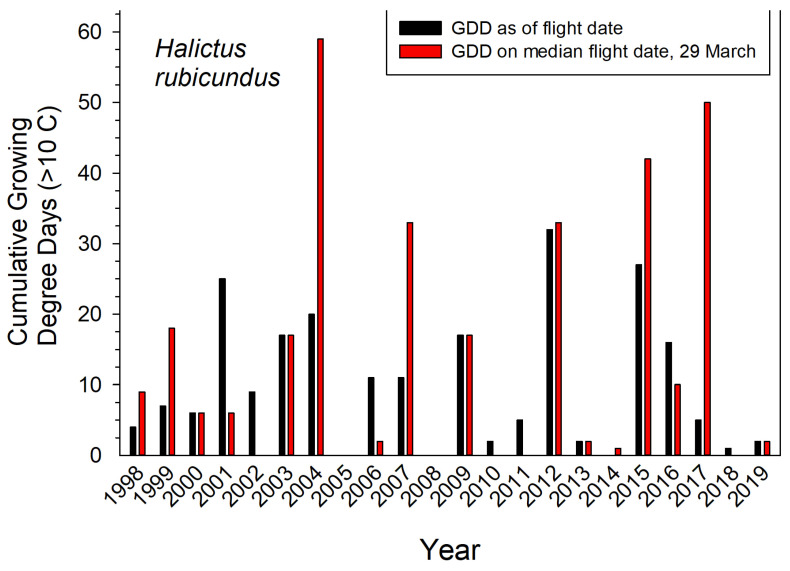
Cumulative growing degree days (GDD > 10 °C) for the actual and median dates that *H. rubicundus* emerged in Logan Utah USA. Temperatures recorded at the campus weather station.

**Figure 4 insects-12-00457-f004:**
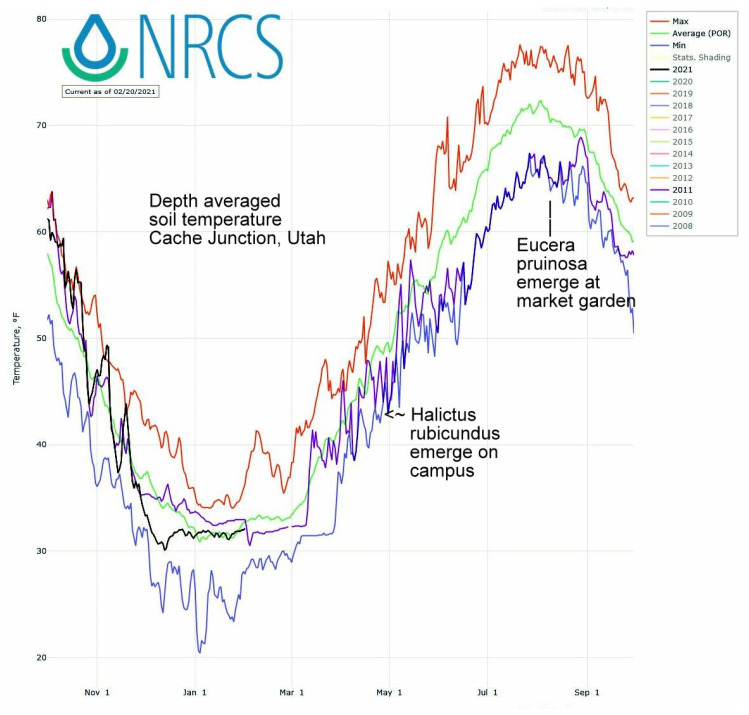
Daily soil temperatures from 20-cm depth at Cache Junction, Utah, across the valley from the market garden with squashes visited by *E. pruinosa*. The plot highlights the daily average soil temperatures over years (green line), daily maxima (red) and minima (blue), and the particularly cold year of 2011 (in violet) when both bee species were late to emerge.

**Figure 5 insects-12-00457-f005:**
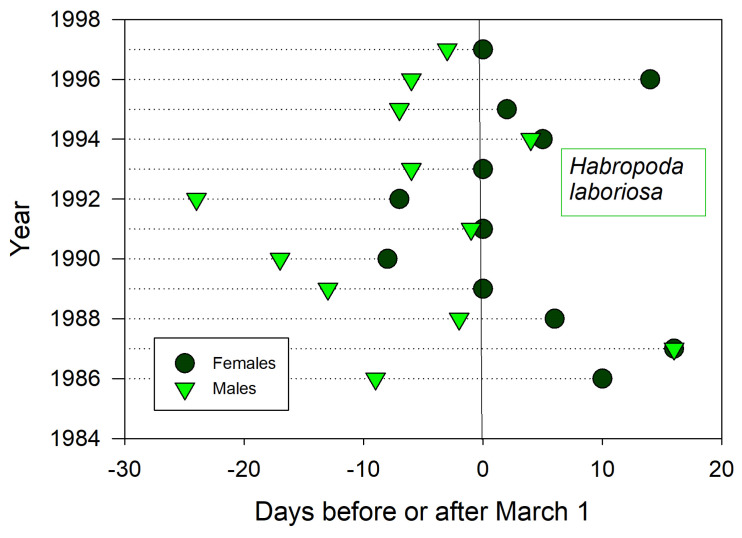
Annual date for first sightings of male and female *H. laboriosa* at flowering blueberries *(Vaccinium)* near Auburn, Alabama, USA.

**Figure 6 insects-12-00457-f006:**
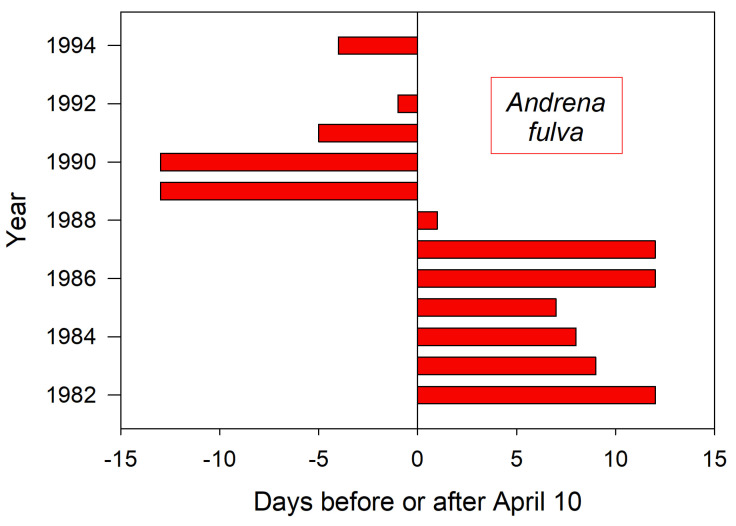
Annual variation in the emergence date for *A. fulva* at their nesting aggregation in Poznan Poland (adapted from data in [13]). Data is missing for 1993.

**Table 1 insects-12-00457-t001:** Bee species studied, their attributes, and the range in days for their first annual flight.

Bee Species	Location	Flight Season	Sociality	Floral Assoc.	First Event Recorded	Years Monitored	Range for 1st Date of Activity (Days)
*Halictus rubicundus*	Logan, UT, USA	March–September	social	polylectic	nest start	22	44
*E. (Peponapis) pruinosa*	North Logan, UT, USA	July–September	solitary	*Cucurbita* oligolege	floral visit	14	34
*Habropoda laboriosa*	Auburn, AL, USA	March–April	solitary	*Vaccinium* oligolege	floral visit	12	24♀, 40♂
*Andrena fulva*	Poznan, Poland	April–May	solitary	polylectic	emergence	13	25

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
