# Peer review of "Global Warming, Advancing Bloom and Evidence for Pollinator Plasticity from Long-Term Bee Emergence Monitoring"

_insects, 2021, doi:10.3390/insects12050457_

Round 1
Reviewer 1 Report
There is interesting and valuable data in the paper on “Global warming, advancing bloom, and evidence for pollinator plasticity from long-term bee emergence monitoring”. The evidence is convincing that emergence is highly plastic among the three species the author observed. The observations from another study on Andrena fulva also suggest considerable plasticity. However, the paper badly needs more organization, and needs to back up many of the statements with references, or remove the unsubstantiated statements altogether. Preferably, the author will take the time to look up and cite references.
The title could be shortened and be more accurate: Evidence for pollinator plasticity from long-term bee monitoring. The paper is not about global warms or advancing bloom per se
An important addition would be a table or two in the methods that states for each species the location, years studied (2002-2014, for example), life history traits (social, solitary, polylectic, oligolectic, etc.) where observations were made (nest site, flowers), what type of temperature data were taken (nearby climate gauge, soil sensors). This would help the reader and organize the methods and results section.
Date of emergence is used throughout the paper, but often the actual observation was the first nest established for Halictus rubicundus (if I am reading the paper correctly). Use of the term emergence should be checked throughout the paper and changed as needed. Also not clear is if all observations of Habropoda laboriosa are at flowers or if sometimes the nearby nest aggregation was included in observations.
The Methods introduces the species in the order of H. rubicundus, H. laboriosa, A. fulva, and E. pruinosa. Normally, the results would talk about these species in the same order but instead the order is H. rubicundus, E. pruinosa and H. laboriosa. A. fulva is not part of the results. This is difficult to follow as a reader.
The timing of first activity is compared to long-term studies of plants from the very high elevation Rocky Mountains and in the Hudson Valley New York. This is an understandable choice but then only some of the plant species from those studies are chosen for comparison instead of all with no real justification. More detail is absolutely needed on why those species are particularly good comparisons to the bees studied in this paper.
Many statements need references, should be rephrased, or deleted. Among them are those on lines, 37-42, 57, 222 (of any western US population), 251, 252, 267 (there are other studies that could be cited).
Fig 5 should be changed from Fahrenheit to Celsius, and the legend has a number of years that do not seem to be represented on the figure.
More specific comments:
L42- urban sprawl and global warming are long-running too. Combine with preceding sentence.
L44. Instead of “rare” say exactly how many
L45. Instead of “several” say exactly how many
L48 does “abundantly collected” mean well-represented in collections?
L51-53. Reword for clarity
L55. Are you proposing that less responsive or flexible are alternative hypotheses? If so, state them as such. What are the references for these ideas?
L58. “uncommon” at this point some summary of the three bee studies should be included, or does this refer to plants, or both bees and plants?
L67. Studied is not needed, and “represent” could be replaced by occurred.
L59-62. This type of information is known about many plants. Is the point here that this is typical, or broadly applicable to plants and bees, or what?
L62. Broadly similar thermal strategy is vague
L70, and sociality too, right?
L71. Not clear what compiled weather station data means
L73. More accurately you recorded “Date of first observed adult activity..”
L82. Perhaps put all of the information of Andrena fulva in the discussion and remove it here?
L92. Preceded in bloom
L97-98. Possible rewording? ..aggregation and return to nest the following spring.
L100. Delete, and the recording weather station.
L103. Logan bees should be “the two bee species studied in Logan, Utah”.
L107. be reader-friendly and explain what forsythia is and why this is included.
L118. Explain the assumptions
L.146 promising means ?????
L148. Does H. rubicundus go to the Iris. Why is this plant particularly relevant to the study?
L159. First sighting means observation of first nest, right?
L163. Remove commas
L165. Does emergence mean nest
L164-166. This is unclear.
L172. Delete “in the valley.”
L184-190. Is any of this data from observations at the nest aggregation?
L190. The last sentence can be deleted and this should be stated in the Methods
L196. Change annual emergence schedule to annual date of first activity because of data on H. rubicundus
L198-200. A section on the flowering plants should be added to the Results section
L205. More typical of single academic granting cycles is not needed.
L209—Either add a section on A. fulva to the Results section or rewrite this section after you talk about your findings from the three species from the US
L238-242. Could be worded more clearly
L243-246. This seems tangential as written. Add more context or delete
L253. Remove sentence on fussy respirometer.
L257. The sentences above are about bees at the southern end of their distributions so does alpine bee populations mean bees at the norther edge of their distributions?
L273. Typo “emerges”
L276-277. Is this a problem? The problems are often lack of vouchers it seems.
Reviewer 2 Report
I have now reviewed the manuscript insects-1206096, entitled “Global warming, advancing bloom, and evidence for pollinator plasticity from long-term bee emergence monitoring”. In this manuscript, the author presents long-term monitoring data on the emergence dates (or more accurately, proxies thereof) of four ground-nesting bee species and explore how emergence dates vary with underlying climatic conditions as the world gradually warms over the last few decades. I agree with the author that long-term data enabling examination of natural variability in wild bees’ timing of activity is currently lacking among published studies, and thus the study represents an important contribution to our knowledge of basic bee biology. The manuscript is well written and well communicated, though with some details lacking as outlined in my comments below. I am hopeful that the author can address all of my comments and queries below in an improved manuscript.
Major / general comments:
- Since the author did not actually measure “emergence” but rather metrics that serve as proxies for emergence (e.g., bees seen at flowers, bees initiating nesting behavior), I think the language will need to be a little more precise throughout the manuscript. I believe there also needs to be a more explicit and thorough discussion of why it is acceptable to use these different metrics as reliable proxies for actual emergence (currently such a discussion is present but insufficient in my opinion).
- Although the descriptive findings of this study can stand on their own merit, the author did refer to some statistics (e.g., L117-119, L145-152), which were described only extremely cursorially. Were these multiple regressions? Does the data structure justify the use of slightly more complex models (e.g., condensing climatic variables to PCA axes)? I think it is very necessary for the author to bolster the analysis portion of the manuscript by providing more details about the analyses (including what the independent and dependent variables are, the class of models, etc.) such that the results could be properly interpreted.
- Currently the manuscript concludes with methodological considerations, including ideas about improving future studies of this kind. I would also like to see a conclusion paragraph of sorts that also provides the broader context of how the study contributes to the field of wild bee natural history and conservation—e.g., what do we know now about the field that we did not before, and how does this knowledge transform our understanding of how bees are responding to climate change?
- A large body of work on the impacts of climate change on pollination mutualisms has focused on plant-pollinator mismatches in their responses to shifting climates. The findings of this study are clearly relevant for this topic, but author has provided surprisingly little discussion on this front (other than that similar magnitude of variation in some bees and some plants perhaps bodes well for their synchrony—which does not seem to be backed up by empirical data, which the author may actually have on had?). I would strongly encourage the author to explore this connection more thoroughly.
- I currently do not see a “data availability” statement of where the raw data will be deposited, despite the fact that the journal’s guideline explicitly states that “publication of your manuscript implies that you must make all materials, data, and protocols associated with the publication available to readers.” Please make the data available to the scientific community, in an easily accessible format (i.e., such that future researchers will not have to use an image-analysis software to manually extract the data from the figures).
Minor / specific comments:
L37-43: These may fall under “common knowledge” at this point, but perhaps it would be good to cite at least a few studies to substantiate these claims.
L54-56: Or conversely, bees could be MORE flexible and leave their plants behind?
L58: Please spell out binomial name in full and cite taxonomic authority in the first appearance of the organism (in this case the blueberry).
L73: “annual onset…annually monitored”; slight redundancy in the double-use of “annual”? Also, since there are gaps in the data, perhaps this statement needs to be qualified.
L78-79: For these data to be truly useful, does it necessitate that the bee becomes active AFTER blueberry flowers have begun blooming? Because how else would adults emerging BEFORE blueberry blooms be tracked? Same comment for Eucera pruinosa in L85-86 on flowering squashes.
81: Was this nesting aggregation then monitored?
L106: Capitalize and italicize “Forsythia” since it is also the genus name?
L114: What is Mesowest?
L133-134: Could the H. rubicundus aggregations all become active near the same time because they come from one big source of overwintering population?
Figure 1: Since emergence is not actually explicitly measured, but rather, proxies thereof (first nest establishment or first time observed at a flower), I’m not quite comfortable with the X axis title.
Figure 2: How is April 1 chosen as the reference point for this figure? Is it the mean date of recorded onset of activity (certainly not the median which is April 11)?
Figure 3: Same comment as Figure 2; also, when do the squash flowers start blooming? I think that is an important piece of information to include as well. Also, just wondering—in places where there is no bar, are the bees emerging exactly on July 10? Or does the year lack a data point?
L161-162: I fail to see how April 11 is the median date since Figure 2 shows that most of the emergence occurred even before April 1. Am I missing something here? Does “median” actually refer to the median date of the full range, rather than the median of the population of sampled years (the latter is the default interpretation), and if so, why this unconventional choice? Also, I can see why the author chose to analyze GDD up to April 11 since GDD up to the actual date of first observed activity did not reveal any trends (L164-165), but it still seems to make little sense to take into account the GDD of days AFTER onset of activity if those days have become irrelevant for the bees’ decision-making to break dormancy. I would encourage the author to critically reconsider what is being communicated here, and see if an alternative analysis would be more informative.
186-188: Is it possible at all that any male H. laboriosa precede even the blooms of V. elliottii? This also brings up the point that squash bloom start dates should be reported as well—at least something like “the first E. pruinosa were observed on squash flowers XX-XX days after the first blooming of squash flowers, on average XX days”. I think it is important to establish that these plant observations really can be expected to capture the very first individuals.
Figure 4: I can see what the figure is trying to communicate, but can’t quite grasp the importance of the figure. Since April 11 is later than all but 3 or 4 of the actual dates of onset of activity, doesn’t it naturally mean that the red bar will be much taller than the black bar in the vast majority of comparisons? What is this figure trying to communicate to the reader?
Figure 5: Perhaps because of the low resolution of the figure, I can’t distinguish individual years, only the main trends. I’m once again unsure of exactly what this figure tells us about the emergence trends of bees, though, and wonder if this figure would be better served as a supplement?
L197, 200-201: The bees’ emergence window is expressed in weeks but the forbs’ blooming window is expressed in days. Consider using the same format?
L207-208: Even though there are many fewer data points for E. pruinosa, I think some sort of formal correlation analysis to see if E. pruinosa and H. rubicundus exhibit similar trends to each other in the same year would be instructive.
L246: A typo in “anima ltaxa”.
L249: Some grammatical issues with “in to climate” here…
Reviewer 3 Report
The paper is a rare example of long term observations of bee emergence in nature and as so, it is a valuable source of information.
I only have a few minor question/corrections to the text and Figures in this paper:
In the Materials and Methods section can you add more information on bee`s biology and on what plants were they observed. Most of this information is there, but not for all. Preferable a table with the most basic information can be add (bees species, location flying period, poly/oligolectic, social/solitary species, most visited/preferred flowers, flowers used in the phenological analysis additionally the measured used for predicting annual onset of bee`s activity)
Fig. 1. is somewhat difficult to understand, I usually prefer Figures as they help to understand the presented data, in this case I would recommend a Table form.
Fig. 5. Temperature on the y-axis would be better in Celsius.
Can you add the correlational data of GDD and emergence date for all bees on a Figure?
Author Response
Response to Reviewer 2
I like the reviewer’s idea for a table. I have added this near the Materials and Methods (even though it contains results as well), and will delete Figure 1 after consulting with the Handling Editor, as its data is now in the table as well.
Figure 5 is generated from the interactive interface for the weather station web site. It does not offer control over the temperature scale. The scale itself is irrelevant for the intended interpretation of this graph, which is to show the range in soil temperatures and latest emergence of both a spring and a summer bee in the years with the coldest soil temperatures.
Correlational data for GDD needs the attending text description for interpretation, as some are for air temperatures, others for soil temperatures, different base temperatures are used for spring and summer bees, and the date chosen for the measure differs with the bee species.
Round 2
Reviewer 1 Report
It reads much better. I have no further comments. Nice contribution!